# Dietary Curdlan Enhances Bifidobacteria and Reduces Intestinal Inflammation in Mice

**DOI:** 10.3390/nu13041305

**Published:** 2021-04-15

**Authors:** Shafaque Rahman, Mark Davids, Patricia H. P. van Hamersveld, Olaf Welting, Hakim Rahaoui, Frank Schuren, Sybren L. Meijer, René M. van den Wijngaard, Theodorus B. M. Hakvoort, Wouter J. de Jonge, Sigrid E. M. Heinsbroek

**Affiliations:** 1Tytgat Institute for Liver and Intestinal Research, Amsterdam Gastroenterology Endocrinology and Metabolism, Amsterdam University Medical Centers, Location AMC, University of Amsterdam, 1105 BK Amsterdam, The Netherlands; s.rahman@amsterdamumc.nl (S.R.); h.p.vanhamersveld@amsterdamumc.nl (P.H.P.v.H.); o.welting@amsterdamumc.nl (O.W.); r.vandenwijngaard@amsterdamumc.nl (R.M.v.d.W.); t.hakvoort@amsterdamumc.nl (T.B.M.H.); w.j.dejonge@amsterdamumc.nl (W.J.d.J.); 2Department of Vascular Medicine, Amsterdam University Medical Centers, Location AMC, University of Amsterdam, 1105 AZ Amsterdam, The Netherlands; m.davids@amsterdamumc.nl; 3The Netherlands Organization for Applied Scientific Research (TNO), 3704 HE Zeist, The Netherlands; hakim.rahaoui@tno.nl (H.R.); frank.schuren@tno.nl (F.S.); 4Department of Pathology, Amsterdam University Medical Centers, Location AMC, University of Amsterdam, 1105 AZ Amsterdam, The Netherlands; s.l.meijer@amsterdamumc.nl; 5Department of Surgery, University of Bonn, 53113 Bonn, Germany

**Keywords:** curdlan, microbiota, intestinal inflammation

## Abstract

β-glucan consumption is known for its beneficial health effects, but the mode of action is unclear. While humans and mice lack the required enzymes to digest β-glucans, certain intestinal microbes can digest β-glucans, triggering gut microbial changes. Curdlan, a particulate β-glucan isolated from *Alcaligenes faecalis*, is used as a food additive. In this study we determined the effect of curdlan intake in mice on the intestinal microbiota and dextran sodium sulfate (DSS)-induced intestinal inflammation. The effect of curdlan on the human intestinal microbiota was assessed using i-screen, an assay for studying anaerobic microbial interactions. Mice received oral gavage with vehicle or curdlan for 14 days followed by DSS for 7 days. The curdlan-fed group showed reduced weight loss and colonic inflammation compared to the vehicle-fed group. Curdlan intake did not induce general microbiota community changes, although a specific *Bifidobacterium*, closely related to *Bifidobacterium choerinum*, was observed to be 10- to 100-fold more prevalent in the curdlan-fed group under control and colitis conditions, respectively. When tested in i-screen, curdlan induced a global change in the microbial composition of the healthy intestinal microbiota from a human. Overall, these results suggest that dietary curdlan induces microbiota changes that could reduce intestinal inflammation.

## 1. Introduction

β-glucans are naturally occurring polysaccharides that form the structural component of microbial and plant cell walls. Depending on their origin, methods of extraction and purification, these glucans have different sizes and are composed of β-1,3, 1,4 and 1,6 glucose linkages [1].

Curdlan is a β-glucan isolated from the cell wall of the bacterium *Alcaligenes faecalis*. It is a linear, water-insoluble (1 → 3) particulate β-glucan, and has been studied for its potential functional roles as a food additive [2] and a sustained-release suppository [3], applications in pharmacology where the sulfoalkyl derivatives of curdlan are used as an anticoagulant, and an HIV inhibitor [4,5]. In 1996, the US FDA approved the use of curdlan in foods to improve texture, stability and thickness [6]. Although thought to be safe for consumption, curdlan is also known to induce inflammatory reactions on a variety of immune cells via Toll/interleukin-1 receptor-like (TIR) domain-containing receptors and dectin-1 [7,8,9,10,11,12]. Due to this ability to activate leukocytes and induce inflammatory responses, curdlan is suggested to boost the immune system’s infection-fighting response and anti-tumor responses [5,13,14,15].

While humans lack the required enzymes to digest β-glucans, certain intestinal microbes can digest curdlan, and some studies have suggested that curdlan has prebiotic potential [16,17,18]. Curdlan was shown to significantly enhance the Lactobacillus, Bifidobacterium and Bacteroides genera in in vitro batch fermentation using human fecal extract [16]. Another study in which curdlan was fed to rats showed an increased abundance of bifidobacteria and short chain fatty acid concentrations in the cecum of the curdlan-fed rats compared to cellulose-fed rats [19].

Gut microbiota dysbiosis has been widely reported and globally recognized in inflammatory bowel diseases (IBD). Reduced bacterial diversity and richness is observed in IBD patients compared to healthy people [20,21,22,23,24,25]. Moreover, many microbial strains act as a probiotic and have been studied in the context of IBD. For instance, *Faecalibacterium prausnitzii* was shown to be protective against IBD in humans [26], and has been shown to induce IL-10 in both mouse and human dendritic cells [27]. Additionally, a specific strain of commensal bacteria, *Bifidobacterium longum,* prevented acute DSS colitis by improving epithelial barrier function in mice [28].

Little is known about the effect of dietary curdlan on the gut microbiota composition and intestinal inflammation. Therefore, we aimed to investigate how curdlan feeding affects innate immune cells, microbial composition, and the course of DSS-induced intestinal inflammation in mice. We also determined the effect of curdlan on the microbial ecology of healthy human fecal microbiota through an in vitro microbiome interactions platform (i-screen) [29]. We established that curdlan affects human and mouse microbial composition, and that in mice, two weeks of dietary curdlan reduces signs of inflammation in DSS-induced colitis. Together, this study shows that the food additive curdlan may affect gastrointestinal health.

## 2. Materials and Methods

### 2.1. Mice

Mice that were 10 to 14 weeks old were used for all experiments. Female C57BL/6 mice (Charles River laboratories, Maastricht, The Netherlands) were used for the DSS experiment and were acclimatized in the Animal Research Institute AMC (ARIA). In-house bred C57BL/6 male and female mice were used for the isolation of peritoneal macrophages. Mice were kept under specific pathogen-free conditions in the ARIA and were maintained on a 12/12 h light/dark cycle in 20  ±  2 °C and 55% humidity with ad libitum food and water access. Mice were randomly distributed between experimental groups and were kept and handled in accordance with the guidelines of the Animal Research Ethics Committee of the University of Amsterdam. All mouse experiments were approved by the Animal Research Ethics committee under code AVD1180020172124.

### 2.2. DSS-Induced Colitis Experiment

The total sample size was 30 mice, divided over 4 groups (vehicle 5 mice, curdlan 5 mice, vehicle+DSS 10 mice, curdlan+DSS 10 mice). Oral gavage was given daily for 14 days with 10 mg/mL of curdlan (Sigma-Aldrich Chemie BV, Zwijndrecht, The Netherlands) in vehicle (100 μL per mouse) or vehicle only (5% (*w*/*v*) glucose (Baxter Healthcare Ltd., Norfolk, UK)). After 14 days, colitis was induced with 2% (*w*/*v*) DSS (TdB Consultancy, Uppsala, Sweden) added to the standard drinking water (Innovive, Thermo Fisher Scientific, Leiden, The Netherlands) and refreshed daily for 7 days. Afterwards, mice received standard drinking water for two days. Body weights were recorded daily during this period (9 days). Control mice were treated with curdlan in vehicle or with vehicle only for 14 days, after which they received standard drinking water until the end of the experiment. At the end of the experiment, mice were euthanized with CO_2,_ and organs were collected. The wet weights of colons together with the total colon length were measured. Colon weight per length was used as a disease parameter. The scoring of colon contents was done according to previous studies [30,31]. Disease activity index scoring was based on the sum of the weight loss score, the stool consistency score and the stool bleeding parameters [32]. One half of the longitudinally divided colons was used for histology and the other half was snap-frozen for RNA isolation.

### 2.3. Histology

Longitudinally divided colons were rolled, fixed in 10% (*w*/*v*) paraformaldehyde for 24 h and transferred to 70% (*v*/*v*) ethanol for routine histology [30,31]. An experienced pathologist evaluated formalin-fixed hematoxylin and eosin-stained tissue sections microscopically, in a randomized blinded fashion. Intestinal inflammation was scored based on eight characteristics of inflammation as described by Heinsbroek et al. [33]. This resulted in a total histology score ranging from 0 to 32 [30,31,33]. Representative pictures of hematoxylin and eosin-stained tissue colon tissue sections were taken using an Olympus BX51 (Olympus Nederland BV, Leiderdorp, The Netherlands).

### 2.4. Isolation and Culture of Peritoneal Macrophages (PM) in Ex Vivo Set Up

Mice were euthanized with CO_2_ after a two-week period of daily oral gavages with curdlan (10 mg/mL) or vehicle. Peritoneal macrophages were isolated by flushing peritoneum with sterile PBS (Fresenius Kabi, Huis Ter Heide, the Netherlands) and 5 mM UltraPure^TM^ EDTA (Fisher Emergo B.V, Landsmeer, The Netherlands). Cells were plated at a concentration of 2 million cells/mL on a 24-well tissue culture plate (VWR International BV, Amsterdam, The Netherlands), stimulated with RPMI-1640 medium, which contains L-glutamine and 25 mM HEPES (Thermo Fisher Scientific, Bleiswijk, The Netherlands). We also added the supplements of 10% (*v*/*v*) fetal bovine serum (FBS) (Bodinco BV, Alkmaar, The Netherlands) and 1% (*v*/*v*) penicillin–streptomycin (Fisher Emergo B.V, Landsmeer, The Netherlands). Cells were incubated for 24 h at 37 °C with 5% CO_2_ to let the peritoneal macrophages adhere. Subsequently, the medium was aspirated and fresh medium, or fresh medium with 10 ng/mL lipopolysachcharide (LPS) (Sigma-Aldrich Chemie BV, Zwijndrecht, The Netherlands), was added to the macrophages for 24 h. Thereafter, macrophages were lysed in Tripure isolation reagent (Sigma-Aldrich Chemie BV, Zwijndrecht, The Netherlands) for RNA extraction and the supernatants were stored to measure cytokines.

### 2.5. mRNA Extraction, cDNA Synthesis and qPCR Analysis

Colonic tissues were homogenized for 15 min in Tripure isolation reagent (Sigma-Aldrich Chemie BV, Zwijndrecht, The Netherlands) with stainless steel beads (Qiagen Benelux B.V, Venlo, The Netherlands) using a tissue homogenizer (Qiagen Benelux B.V., Venlo, The Netherlands). After homogenization, mRNA was isolated according to the manufacturer’s instructions. RNA was cleaned up using the Bioline ISOLATE II mini kit (GC Biotech, Alphen aan den Rijn, The Netherlands) in accordance with the manufacturer’s protocol and as described previously [34,35]. RNA from peritoneal macrophages was isolated using the Tripure isolation reagent in accordance with the manufacturer’s protocol and as described previously [34,35]. RNA from both colonic tissues and peritoneal macrophages was used to synthesize cDNA using 0.19 µg/µL of Random Hexamer primers (Promega, Leiden, The Netherlands) and 10 µM Oligo dT primers (Thermo Fisher Scientific, Bleiswijk, The Netherlands) at 70 °C for 5 min, followed by the addition of 1 mM deoxyribonucleotide triphosphate (dNTPs) (Thermo Fisher Scientific, Bleiswijk, The Netherlands), 1× RT-buffer (Thermo Fisher Scientific, Bleiswijk, The Netherlands), 5 U/µL Revertaid Transcriptase (Thermo Fisher Scientific, Bleiswijk, The Netherlands) and 1 U/µL Ribolock RNAse Inhibitor (Thermo Fisher Scientific, Bleiswijk, The Netherlands) for 60 min at 42 °C and 10 min at 70 °C. Quantitative polymerase chain reaction (qPCR) was performed on a LightCycler 480 II (Roche Diagnostic Nederland BV, Almere, The Netherlands) or Bio-Rad CFX96 (Bio-Rad Laboratories BV, Lunteren, The Netherlands) using SensiFAST SYBR No-ROX (GC biotech BV, Waddinxveen, The Netherlands). mTNF-α, mIL-6, mIL-10, mNOS2 and mCCL-2 were measured and analyzed using LinRegPCR software [36]. Glyceraldehyde-3-phosphate dehydrogenase (GAPDH) and Cyclophilin were used as housekeeping genes for colon samples, whereas GAPDH and ribosomal protein, large, P0 (RPLP0) were used for peritoneal macrophages (PM). All the housekeeping genes, TNF-α, IL-6, and IL-10 primers were synthesized by Sigma-Aldrich Chemie BV, Zwijndrecht, The Netherlands, and the NOS2 and CCL-2 primers were synthesized by Life Technologies Europe, Bleiswijk, The Netherlands. The genes and nucleotide sequences are listed in Table 1.

### 2.6. Measurement of Cytokine Protein Concentrations

The TNF-α, IL-6 and IL-10 concentrations were determined in the supernatant of peritoneal macrophages via an enzyme-linked immunosorbent assay (ELISA; R&D systems, Abingdon, UK) following the manufacturer’s protocol.

### 2.7. Human i-Screen

An i-screen was performed with a pooled fecal inoculum of 6 healthy adult human volunteers. Fecal samples were collected anonymously in accordance with The Netherlands Organization for Applied Scientific Research (TNO) operational procedures and was approved by an internal ethical evaluation committee. All participants provided their written informed consent. The exact procedure was as described in Ladirat et al. [37]. Different concentrations of curdlan were mixed with simulated ileal efflux medium (SIEM) and 2% (*v*/*v*) fecal inoculum in each well of a deep-well plate (Thermo Fisher Scientific, Bleiswijk, The Netherlands). Three concentrations (1 mg/mL, 2 mg/mL and 4 mg/mL) were tested in triplicate. Inoculated SIEM without curdlan was used as a control and was included in triplicate. Collected samples were directly stored at −20 °C for DNA isolation after 24 h of fermentation.

Total DNA from the collected samples was isolated as described previously [37] with some changes. At first, the samples were mixed with 300 μL of lysis buffer (Agowa, Berlin, Germany), 500 μL of zirconium beads (0.1 mm) (Lab Services, Breda, The Netherlands), and 500 μL of phenol (Sigma-Aldrich, Taufkirchen, Germany), then the samples were placed in a BeadBeater (BioSpec Products, Bartlesville, OK, USA) for a duration of 3 min. The DNA concentration was measured via NanoDrop 1000 Spectrophotmeter (Thermo Fisher Scientific, Bleiswijk, The Netherlands).

#### Quantification of Total DNA (qPCR)

Quantification of the total amount of DNA present in each sample was performed using universal primers 16S-uni-I-F (10 pmol μL^−1^) (CGAAAGCGTGGGGAGCAAA) and 16S-uni-I-R (10 pmol μL^−1^) (GTTCGTACTCCCCAGGCGG), and the probe 16S-uni-I (5 μM) (6FAM-ATTAGATACCCTGGTAGTCCA-MGB) (Life Technology, Bleiswijk, The Netherlands). Amplification was performed with a 5 μL DNA sample and a 20 μL q-PCR mixture that contained 12.5 μL Diagenode Master Mix (Diagenode, Seraing, Belgium), 1 μL each of the primers and probe, and 4.5 μL of MilliQ water. The total microbial fecal DNAs were diluted (1:50) before use in the q-PCR assay. The PCRs were performed using the 7500 Fast-Real Time PCR system (Applied Biosystems, (Foster City, CA, USA) with the following sequential settings: 1 step of 2 min at 50 °C, 1 step of 10 min at 95 °C, followed by 40 cycles of 15 s at 95 °C and 1 min at 60 °C.

### 2.8. DNA Isolation for Microbiome and Mycobiome Sequencing of Mice Colon Contents

DNA was extracted using mechanical lysis and the PSP Spin Stool DNA Plus Kit (Isogen LifeScience, Utrecht, The Netherlands). Samples were placed in Lysing Matrix E tubes (Bio-Connect, Huissen, The Netherlands) with 1400 µL of stool stabilizer from the PSP kit. Mechanical lysis was performed using FastPrep (BioSPX, Abcoude, The Netherlands) with three repetitive rounds of 30 s at 6.5 m/s, and with cooling for 30 s on ice in between. After the FastPrep procedure, the samples were kept at 95 °C for 15 min, followed by cooling on ice for 1 min and centrifugation at 13,400× g for 1 min. The supernatant was transferred to the PSP InviAdsorb, following the manufacturer’s protocol. Negative extraction controls (DNA-free water) were processed in the same way. DNA concentrations were measured with the Nanodrop spectrophotometer (Thermo Fisher Scientific, Bleiswijk, The Netherlands) and Qubit fluorometric quantitation method (Thermo Fisher Diagnostics, Nieuwegein, The Netherlands).

### 2.9. Bacterial Profiling from Mouse Colon Contents and Human i-Screen-Derived DNA Amplicons

The 16S rRNA gene amplicons were produced using a single-step PCR procedure targeting the V3–V4 region, and this was carried out at the Microbiota Center Amsterdam (MiCA) as described earlier [38]. Amplified sequence variants (ASVs) were inferred for each sample individually with a minimum abundance of 4 reads [39]. Unfiltered reads were mapped against the collective ASV set to establish the abundances. Taxonomy was assigned using the IDTaxa [40] and SILVA 16S ribosomal database V132 [41]. To assess the phylogenetic placement of relevant bifidobacterial lineages, a phylogentic tree was constructed with phangorn V2.5.5 [42], using ASV and bifidobacterial DSM and ATCC strain sequences obtained from the SILVA V132 database [43]. Data obtained from the i-screen samples were processed similarly.

### 2.10. Fungal Profiling from Mouse Colon Contents

The fungal diversity was determined by ITS1 amplicon sequence analysis carried out at the MiCA, as described earlier [38]. PCR-generated amplicon libraries were obtained from 100 ng fecal DNA using the ITS1 primer set containing an overhang for the Illumina Nextera platform.

UNOISE was used to infer amplified sequence variants (ASVs) for each sample individually with a minimum abundance of 4 reads [39]. Unfiltered reads were then mapped against the collective ASV set to establish the abundances. Taxonomy was assigned using the UNITE database [44].

### 2.11. Statistical Analyses

Statistical analysis was done using Graphpad Prism 8.0 (GraphPad Software, La Jolla, CA, USA). All data are expressed in mean ± SEM. Individual *t*-tests were used to evaluate the statistical significance of the difference, whereby *p* was ≤0.05 (*), 0.01 (**) or 0.001 (***).

Microbiome data from the mice colon contents and human fecal DNA from the i-screen were analyzed and visualized in R (V3.6.3) [45]. Bacterial and fungal count tables were rarefied to 10,000 counts before analysis. Phyloseq [46] and Picante [47] were used to calculate alpha diversity metrics, which were tested using Wilcoxon rank-sum testing and linear models. Permutation multivariate analysis of variance (MANOVA) from Vegan [48] was used to test compositional differences. The differential taxa abundance was tested using the DESeq2 package [49], and FDR-corrected *p*-values less than 0.01 were regarded as significant. Linear models were used to assess the dose response of curdlan on specific ASVs.

## 3. Results

Since curdlan is used in the food industry and is thought to activate immune cells and change microbial composition, we aimed to study the effect of curdlan intake on the innate immune system, intestinal inflammation, and microbial composition.

### 3.1. Curdlan Intake Reduces Signs of Inflammation in Acute DSS-Induced Colitis in Mice

To investigate the effect of curdlan intake on intestinal inflammation, WT mice received daily oral gavage with either 1 mg/0.1 mL curdlan or vehicle for 14 days, followed by the induction of DSS colitis for 7 days (Figure 1a). As expected from this model [50], we observed weight loss in the groups receiving DSS (vehicle and curdlan). In the groups receiving DSS, curdlan-fed mice lost less weight compared to vehicle-fed mice at day 9 (*p*-value = 0.06) (Figure 1b). No significant differences were observed in colon weights per cm (Figure 1c) or disease activity index between vehicle and curdlan-fed groups (Figure 1d). The histological scoring of colon pathology suggests inflammation was ameliorated more profoundly in curdlan-fed mice compared to vehicle-fed mice (Figure 1e) in terms of ulceration and crypt loss. A representative histology picture of the colons of vehicle- and curdlan-fed mice under DSS conditions is shown in Figure 1f, which shows less inflammation in the curdlan-fed mice comparatively.

Colonic cytokine levels were also determined and the groups not receiving DSS had low levels of cytokines. Surprisingly, colonic TNF-α levels were increased in curdlan-fed mice compared to vehicle-fed mice under non-colitis conditions (Figure 1g). In the DSS-treated groups, colonic cytokine levels went up, as expected during inflammation, but no significant difference was found between curdlan-fed mice and vehicle-fed mice in terms of TNF-α (*p*-value = 0.22), IL-6 (*p*-value = 0.21), IL-10 (*p*-value = 0.09) or CCL-2 (*p*-value = 0.4) expression (Figure 1g). Together, our data suggest that curdlan feeding reduces signs of intestinal inflammation, including weight loss, ulceration and cryptloss, in acute DSS-induced colitis.

### 3.2. Two Weeks of Dietary Curdlan Reduces LPS Responses in Peritoneal Macrophages

To determine if curdlan intake affects peritoneal macrophage responses, mice were fed with 1 mg curdlan or vehicle via oral gavage for 14 days. At this dose, β-glucans have been found in the circulation in previous studies [51]. At the end of this two-week period, peritoneal macrophages were isolated and stimulated with LPS for 24 h (Figure 2a). Unstimulated macrophages isolated from curdlan-fed mice showed a downregulation of IL-10 protein compared to macrophages isolated from vehicle-fed control mice (Figure 2b). Upon LPS stimulation, a significant downregulation of TNF-α, IL-6 and IL-10 proteins (Figure 2b) was observed in macrophages from curdlan-fed mice compared to macrophages from the vehicle-fed mice. The mRNA levels of these macrophages also showed that curdlan mediated the downregulation of IL-6 upon LPS stimulation, while no effect was seen on the mRNA levels of TNF-α and IL-10. This interesting discrepancy between RNA levels and protein levels suggests that curdlan may affect the post-translational modifications and stability of these cytokines (Figure 2c). These data indicate that curdlan intake may modulate macrophage immune responses.

### 3.3. Mouse Colonic Microbial Diversity Is Affected by Dietary Curdlan

Since curdlan is suggested to have prebiotic potential [16,17,18], we analyzed the impact of dietary curdlan on colonic microbial composition using 16S rRNA and ITS1 sequencing for bacteria and fungi, respectively. The colon contents of mice fed with curdlan or vehicle, with and without DSS treatment, were analyzed. Unfortunately, five samples were dropped from the fungal analysis due to insufficient ITS amplicon yield. As known from other microbiota studies [52,53,54,55,56], DSS treatment has a significant effect on bacterial and fungal alpha (Figure 3a,c) and beta diversity metrics (Figure 3b,d). The number of bacterial observed species was decreased upon DSS treatment (*p* < 0.01) (Figure 3a). Reduced bacterial diversity was also seen via the Shannon index (*p* < 0.05) and Faith’s phylogenetic diversity (FPD) (*p* < 0.001). The effect of DSS on the fungal composition was less pronounced, although the number of observed species was also decreased upon DSS treatment (*p* < 0.05) (Figure 3c).

Dietary curdlan affected microbial diversity in the DSS-treated groups. Bacterial diversity was increased in terms of the number of observed species and the phylogenetic diversity (*p* < 0.05). In contrast, fungal richness was decreased by dietary curdlan (*p* < 0.01). The permutation ANOVA of Bray–Curtis dissimilarity showed that both the bacterial (*p* < 0.001; R^2^ = 0.46) (Figure 3b) and the fungal (*p* < 0.05; R^2^ = 0.25) composition were significantly altered (Figure 3d). Curdlan seemed to induce a slight microbial shift in DSS-treated mice, but no significant effect of curdlan was observed (*p* = 0.15; R^2^ = 0.03 and *p* = 0.20; R^2^ = 0.06, respectively).

### 3.4. Bifidobacterium Abundantly Present after Curdlan Feeding in Mice

DESeq2 differential abundance analysis was used on these samples to test the effects of curdlan on specific clades. As expected, many (88) bacterial clades changed in relative abundance after DSS treatment (Appendix A). The differential abundances of five bacterial ASVs were affected by curdlan feeding irrespective of DSS treatment (Figure 4), two unclassified lineages, a *Lachnospiraceae*, a *Blautia* and a *Bifidobacterium*. Of these, the *Bifidobacterium* genus stood out as the most interesting due to its known beneficial effects on intestinal inflammation [57,58,59,60] and its described ability to enhance proliferation upon curdlan treatment [19]. The *Bifidobacterium* genus was more abundant in curdlan-fed mice, and remained more abundant through DSS treatment. To get a better functional understanding of this possible probiotic genus, the marker sequence was aligned with other relevant bifidobacteria sequences and a phylogenetic tree was constructed (Figure 6). This indicated that this *Bifidobacterium* was most closely related to *Bifidobacterium choerinum*, a species known for its starch-degrading properties and potential probiotic role [61].

### 3.5. Curdlan Additionally, Affects Human Microbial Composition

To determine the effect of curdlan on human intestinal microbial composition, different concentrations of curdlan (0 mg/mL, 1 mg/mL, 2 mg/mL and 4 mg/mL) were used in the i-screen method. In this system, intestinal microbiota are cultured under anaerobic conditions for 24 h; this has been used in the past to illustrate gut microbiota modulation by prebiotics [29]. We saw a significant increase in the total DNA load of the 16S bacterial composition with the addition of curdlan in the human fecal i-screen, which suggests it supports bacterial growth (Figure 5a). The alpha diversity analysis of 16S rRNA sequencing showed increased numbers of microbial species upon the application of different doses of curdlan treatment (Figure 5b). Similarly, a dose-dependent responseof curdlan was also seen in the beta diversity (Figure 5c), and many bacterial clades changed in relative abundance after curdlan treatment (Appendix A). The linear model analysis identified bacterial ASVs that showed differential abundances upon curdlan treatment, as shown in Figure 5d (top six ASVs shown). Since our mouse data did show that curdlan enhanced bifidobacteria, we looked more closely into the effect of curdlan on *Bifodobacterium* species in this human system. Surprisingly, bifidobacteria were not differentially present upon curdlan treatment. Different bifidobacteria were present; however, only the highest curdlan concentration of 4 mg/mL enhanced their presence (Figure 5e).

### 3.6. Relative Abundance of Bifidobacterium Choerinum in Mice and Not Human

Not surprisingly, the bifidobacteria found in the human samples were completely different from the ones found in the mouse colon. To investigate the effects of curdlan on the specific beneficial bifidobacterial, as seen in mouse colon contents, a phylogentic tree was constructed (Figure 6) using ASVs, and bifidobacteria DSM and ATCC strain sequences obtained from the SILVA (version 132) database [43]. This was done for both sequences obtained from mouse colon contents and the human i-screen. The placement of bifidobacteria clearly shows the different species of the family present in mice and humans, respectively. The abundance of ASV_467 was found to be closely related to *Bifidobacterium choerinum*, which is only limited to mouse colon contents. Even though it is clear that mouse and human intestinal microbiota are not comparable, together these data suggest that curdlan can alter both human and mouse gut microbial composition.

## 4. Discussion

Various health benefits have been assigned to dietary β-glucan intake [5,13,14,15]. Here, we show that dietary curdlan, a particulate β-glucan, alleviates DSS-induced inflammation in mice, and affects intestinal microbiota composition and macrophage LPS responses.

Several studies have investigated the effects of β-glucan consumption on intestinal health [33,62], and a reduction in intestinal DSS-induced colitis has been described following the oral administration of various β-glucans [63,64,65]. DSS-induced colitis is an innate driven acute inflammation of the colon caused by epithelial barrier disruption [66,67]. Reduced inflammation in DSS colitis therefore suggests that innate immune responses are affected by β-glucans. Interestingly, in this study, peritoneal macrophages isolated from mice that received dietary curdlan showed reduced LPS responses compared to macrophages from vehicle-fed mice, as shown in Figure 2. Together, this suggests that curdlan may affect innate immune responses in mice.

The mechanisms via which curdlan reduces innate immune response remain unclear. Possibly, part of the ingested β-glucan enters the system and induces innate immune memory [68,69,70]. However, curdlan-mediated innate immune memory generally leads to innate immune training, which increases the response to a second immune challenge, such as LPS [69,71]. We did not observe innate immune training, and therefore probably other mechanisms led to the observed reduced inflammation.

Several studies have shown that microbial-derived metabolites can directly affect macrophage responses [72,73,74,75]. Furthermore, β-glucans have been shown to induce the growth of lactobacilli and bifidobacteria with probiotic properties [76,77]. Therefore, it is likely that in our colitis experiment, curdlan induced the growth of beneficial microbes, which led to reduced innate immune activation. Indeed, one previous study showed that dietary curdlan increased the presence of bifidobacteria in the cecum of rats [19]. Bifidobacteria are generally beneficial gut bacteria, and a higher abundance of these species is often related to the amelioration of experimental ulcerative colitis [57,78]. Furthermore, *B.animalis subsp. lactis* strain BB12 was shown to have a protective effect on DSS colitis in mice [79]. Additionally, IBD patients display decreased levels of bifidobacteria during active disease [57,80]. Through differential abundance analysis performed on the 16S rRNA sequencing data, we could observe ASV_467 being positively associated with curdlan feeding in both curdlan-fed groups (with and without DSS) compared to vehicle-fed mice (Figure 4, left panel). Based on the sequences obtained, we could plot a phylogenetic *Bifidobacterium* tree, as shown in Figure 6, to assess the placement of the ASVs generated. ASV_467 was seen to be closely related to *Bifidobacterium choerinum. B.choerinum* is thought to be well adjusted to the gut of pre-weaned piglets [81]. This specific *Bifidobacterium,* in particular *B.choerinum* FMB, has non-digestable starch-degrading properties, and is also referred to as a potential probiotic candidate [61,82]. We are limited in our knowledge of the specific bifidobacteria strain that is positively affected by curdlan feeding in our mice, but it is a possible probiotic species that may digest curdlan, and thus give rise to an ecological advantage in the curdlan-fed mice.

In contrast to the curdlan-induced bacterial taxa changes in mice, curdlan’s effect was less pronounced on fungal communities. Our mice had a low colonic fungal load, which led to insufficient ITS1 amplicon yield. However, our data do show that dietary curdlan reduced fungal richness both under healthy and inflammatory conditions (Figure 3c). Furthermore, DSS-induced inflammation decreased fungal richness. Several studies have shown that during inflammation, the intestinal mycobiome changes in IBD patients [83,84] and murine IBD models [85,86], and our data confirm this. However, a low fungal burden made it difficult to further screen the specific taxa influenced by DSS and curdlan.

Moreover, curdlan also induced changes in the human intestinal microbiota. As seen in Figure 5c, a dose-dependent response was observed in the microbiome composition. However, even though the presence of different bifidobacteria species was increased with curdlan treatment of 4 mg/mL (Figure 5e), no specific bifidobacteria had a significant linear relation with the curdlan concentration. Additionally, a correlation with mouse ASV_467 was not found in the screening (Figure 6). This is likely due to differences in intestinal microbiota composition between human and mice.

Crohn’s disease and ulcerative colitis patients are found to have lower levels of *Lachnospiraceae* and *Bacteriodetes* in their colons compared to healthy controls [87,88,89]. Although these bacteria were differentially present after the addition of curdlan into the i-screen (Figure 5d), it is difficult to extrapolate the effect of curdlan treatment on them, as demonstrated by the respectively increasing and decreasing species abundances with increasing concentrations of curdlan. Interestingly, the abundance of certain *Blautia* species was increased with increasing curdlan concentrations. *Blautia* species are associated with beneficial butyrate production [90], and blautia abundance is reduced in IBD patients [91,92]. An increase in *Blautia* species is also associated with improved survival amongst graft-versus-host disease patients [93], and *Blautia* species were associated with remission in ulcerative colitis patients following fecal microbiota transplantation (FMT) [94]. Thus, we see that curdlan affects the overall global microbial change in human as well, but for now, it remains uncertain if this will have a probiotic effect on humans.

In conclusion, our findings show that dietary curdlan can induce intestinal microbiota changes in both human and mice, which may affect intestinal inflammation.

## Figures and Tables

**Figure 1 nutrients-13-01305-f001:**
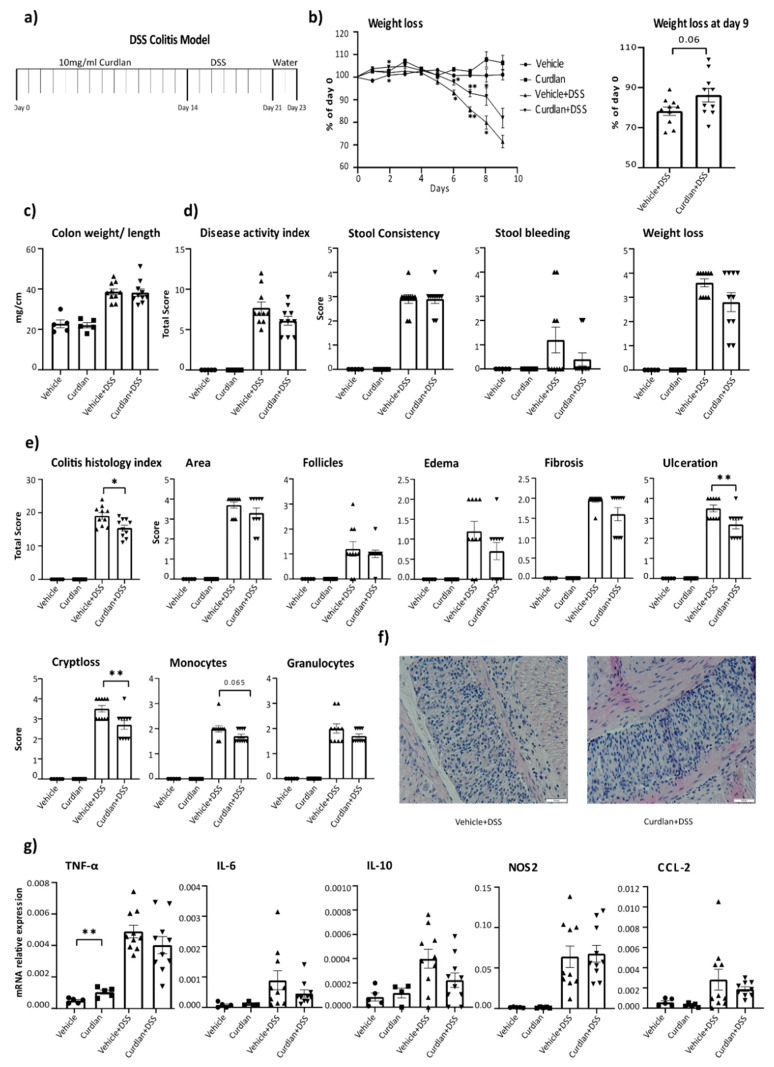
Effects of curdlan intake on dextran sodium sulfate (DSS) colitis in mice. (**a**) A schematic presentation of curdlan feeding in mice. (**b**) bodyweight changes of mice with and without DSS colitis, compared to day 0. (**c**) Colon weight normalized to length as an indication of inflammation. (**d**) Disease activity index according to [32]. (**e**) Mouse colitis histology index according to Table 1 [33]. (**f**) Representative picture of colon histology of vehicle- and curdlan-fed mice with DSS-induced intestinal inflammation, 10× magnification. (**g**) mRNA expression of TNF-α, IL-6, IL-10, NOS2 and CCL-2 in colon homogenates. The levels of mRNA were normalized for reference genes. Samples with no amplification were considered as 0. *N* = 4–5 for controls and 9–10 for DSS-subjected mice. Individual values are expressed as mean and standard error of mean. Statistical differences were tested by independent *t*-test, where a *p*-value of <0.05 was considered to be significant. * *p*-value <0.05; ** *p*-value < 0.01.

**Figure 2 nutrients-13-01305-f002:**
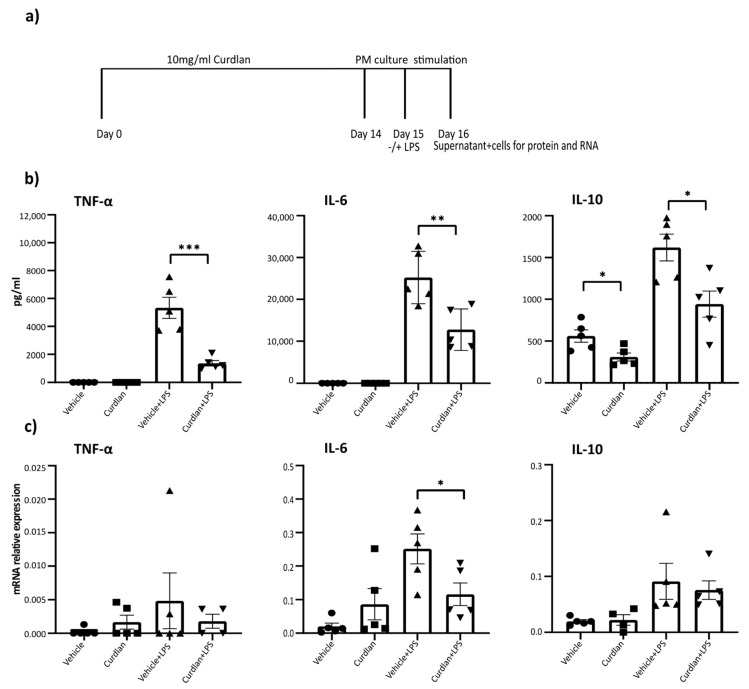
Curdlan intake results in reduced lipopolysachcharide (LPS) responses in peritoneal macrophages. (**a**) A schematic presentation of curdlan feeding in mice and peritoneal macrophage stimulation. (**b**) Cytokine response by peritoneal macrophages after LPS stimulation for 24 h. (**c**) mRNA gene expression levels of TNF-α, IL-6 and IL-10 after 24 h of LPS stimulation. The levels of mRNA were normalized for reference genes. Samples with no amplification and below detection levels were considered as 0. *N* = 3–5 for each condition. Individual values are expressed as mean and standard error of mean. Statistical differences were tested by independent *t*-test, where a *p*-value of < 0.05 was considered to be significant. * *p*-value < 0.05; ** *p*-value < 0.01; *** *p*-value < 0.001.

**Figure 3 nutrients-13-01305-f003:**
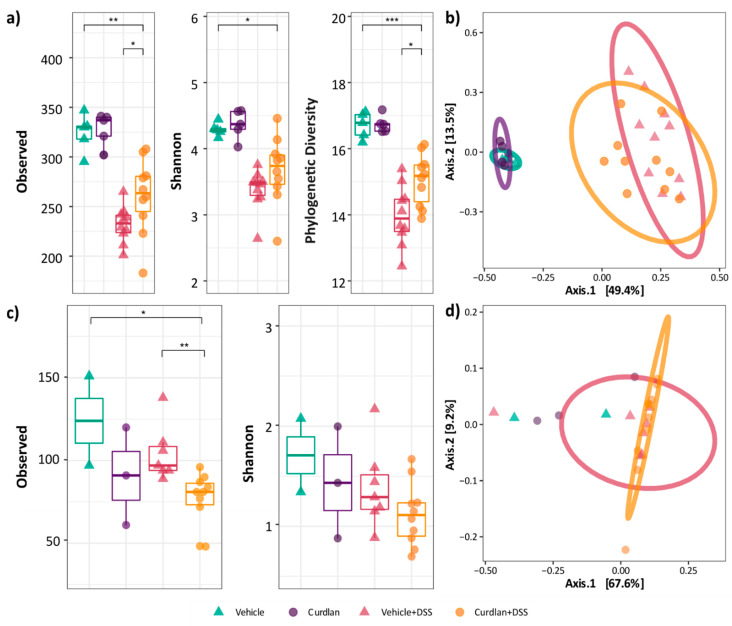
Effect of curdlan and DSS on the microbiome and mycobiome alpha and beta diversity. (**a**) For 16S sequencing, alpha diversity is measured by observed, Shannon and phylogenetic diversity matrices. (**b**) Beta diversity plots for 16S. (**c**) For ITS1 sequencing, alpha diversity is measured with observed and Shannon indices. (**d**) Beta diversity for ITS1 sequencing. Individual samples are represented as symbols. The observed species are plotted for vehicle (green), curdlan (purple), vehicle+DSS (pink) and curdlan+DSS (orange). Statistical differences were tested through Wilcoxon methods, where *p*-value of <0.05 was considered to be significant. * *p*-value < 0.05; ** *p*-value < 0.01; *** *p*-value < 0.001.

**Figure 4 nutrients-13-01305-f004:**
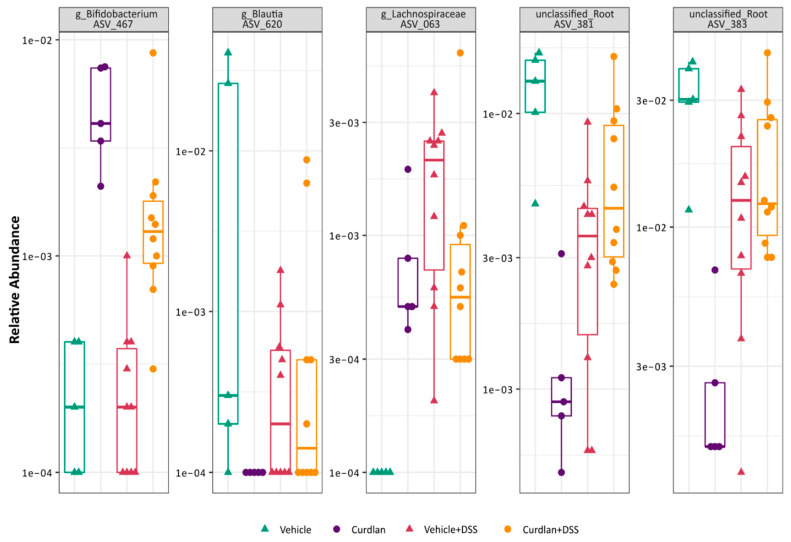
Abundance of bacterial taxa affected by curdlan. Phylogenetic diversity matrices are plotted for the observed species for vehicle (green), curdlan (purple), vehicle+DSS (pink) and curdlan+DSS (orange). Individual samples are represented as symbols. Statistical differences were tested through Wilcoxon methods, where *p* < 0.05 was considered to be significant.

**Figure 5 nutrients-13-01305-f005:**
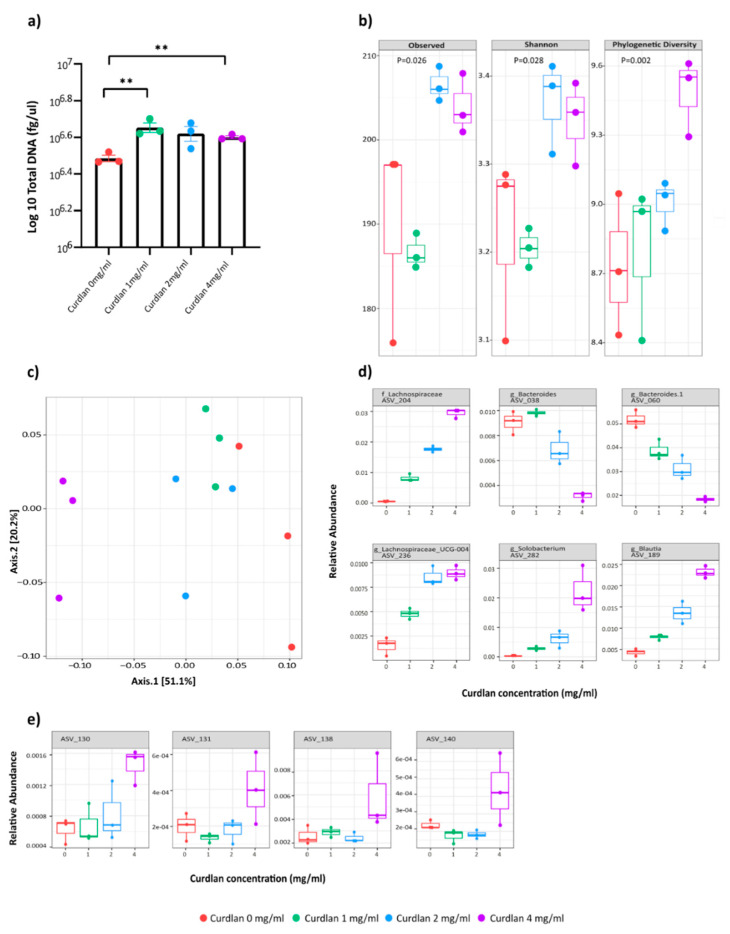
Effect of curdlan on human microbiota using in vitro i-screen. (**a**) Total DNA load of bacterial 16S as measured through qPCR. Statistical differences were tested by independent *t*-tests, where a *p*-value of < 0.05 was considered to be significant. ** *p*-value < 0.01. (**b**) For 16S sequencing, alpha diversity is measured by observed, Shannon and phylogenetic diversity metrics, and significance is tested using Kruskal–Wallis. (**c**) Effect of curdlan concentration on beta diversity (principal coordinate analysis (PCoA) Bray–Curtis dissimilarity)). (**d**) Boxplots for the top 6 most significant lineages (family and genus) that show linear relation with different curdlan concentrations. (**e**) Boxplots for 4 *Bifidobacterium* lineages with different curdlan concentrations.

**Figure 6 nutrients-13-01305-f006:**
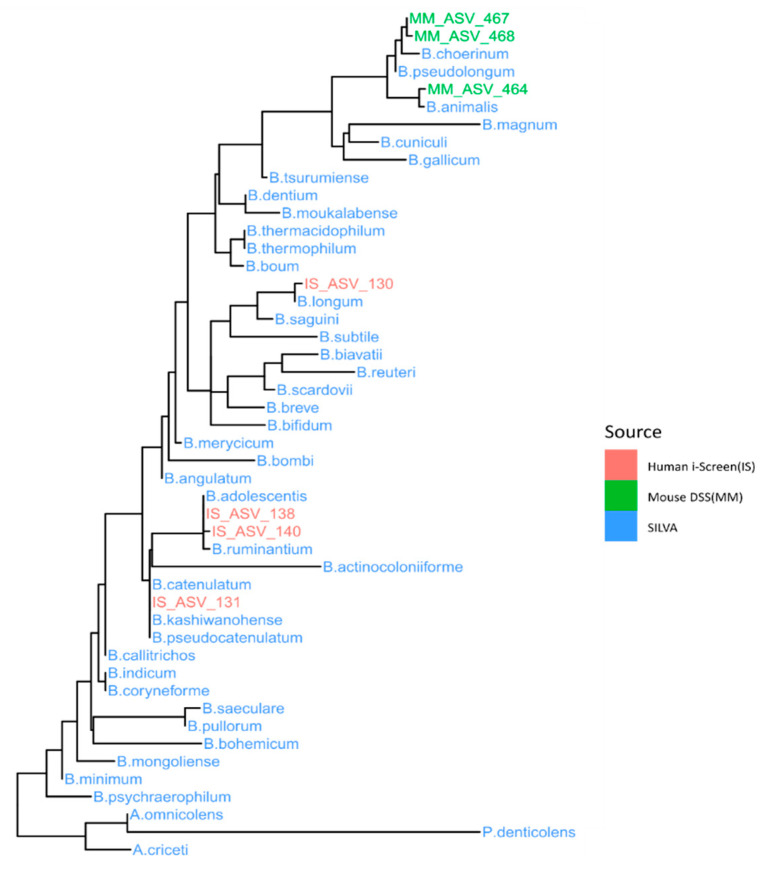
A phylogenetic tree showing the placement of relevant bifidobacteria lineages in mouse and human. The phylogenetic tree shows bifidobacteria lineages in human (pink) and mouse (green) and the reference SILVA database (blue).

**Table 1 nutrients-13-01305-t001:** Primer sequences used in quantitative polymerase chain reaction (qPCR) reactions.

Gene	5′-Forward Sequence	5′-Reverse Sequence
GAPDH	ATGTGTCCGTCGTGGATCTGA	ATGCCTGCTTCACCACCTTCT
Cyclophilin	ATGGTCAACCCCACCGTGT	TTCTGCTGTCTTTGGAACTTTGTC
RPLP0	CCAGCGAGGCCACACTGCTG	ACACTGGCCACGTTGCGGAC
TNF-α	AAAGCATGATCCGCGACGT	TGCAAGCAGGAATGAGAA
IL-6	GAGTTGTGCAATGGCAATTCTG	TGGTAGCATCCATCATTTCTTTGT
IL-10	TGTCAAATTCATTCATGGCCT	ATCGATTTCTCCCCTGTGAA
NOS2	TTCTGTGCTGTCCCAGTGAG	TGAAGAAAACCCCTTGTGCT
CCL-2	AGGTCCCTGTCATGCTTCTG	TCTCCAGCCTACTCATTGGG

## Data Availability

Not applicable.

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
