# Peer review of "Dietary Curdlan Enhances Bifidobacteria and Reduces Intestinal Inflammation in Mice"

_nutrients, 2021, doi:10.3390/nu13041305_

Round 1

Reviewer 1 Report

The manuscript has been imporved.

Reviewer 2 Report

The new version of the manuscript is much improved than the original version with all the suggestions included.

This manuscript is a resubmission of an earlier submission. The following is a list of the peer review reports and author responses from that submission.

Round 1

Reviewer 1 Report

The manuscript is written well. However, I would like to draw your attention to the following suggestions to improve the quality of paper. 

Line 166: Correct CO2.

Origin of all chemicals and equipment should be indicated with company name and the country. Correct throughout the manuscript.

Line 168: use lower case letters to write Lipopolysachcharide.

Line 612: Remove italics of  Bifidobacteria. Check throughout the manuscript.

Reviewer 2 Report

The authors explore the potential functionality of curdlan in inflammation by murine models, evaluating also the impact of the extract on microbiome. However, there are major and minor problems that should be solved.

Major

- English needs to be deeply revised. There are taxnomic mistakes and in some paragraphs it is difficult to be sure abour the context and meaning.

- I do not used it, but what I have read about Tripure isolation reagent is that is recommended for leaf extraction. Had you ever used this kit before for animal samples? Please include bibliography in which the protocol was satisfactorily done. If not, one can assume that this was not the good choice for this work.

-  L183-184. As the step RNA to cDNA is critical, it is mandatory to either explain concentrations and conditions for the reaction, or cited previous publications running the same assay.

- L229: do the human participants sign any informed consent? Was this accepted by ethical committee? if this is the case, please specifically declare it.  If not, maybe it is not possible to publish this part of the work.

- Figure 1g. “curdlan fed mice showed slight but not significantly reduced cytokine levels compared to vehicle fed mice with TNF-α (P-value= 0.22), IL-6 (P-396 value= 0.21), IL-10 (P-value= 0.09) and CCL-2 (P-value= 0.4)” It is not shown any reduction, neither slight. P-values are quite high. Please revise the writing to better explain the results. “non reduction was observed in cytokine levels compared to vehicle fed mice”. Change also the title of the section accordingly.

- Figure 2. Why do you think IL-10 levels change significantly, but not mRNA levels?

-Figure 3, Improve the quality of the figures and please be more accurate in captions (ie 16s rRNA).

- Figure 5. Please split the figure in more, or include some in supplementary. It is hard to read them.

  • Figure 6. The length obtained by V3-V4 sequencing is too short to evaluate at species level. You can use a distance tree, but conclusions might be cautious
  • Please include information about number of sequences obtained, quality, etc. If not, it is not possible to evaluate the robustness of the data and consequently of the results and conclusiones

- L624-625. IN discussion, please change and do not refer to any strain, as 16s rRNA sequencing has not sensitivity to differentiate at strain level (251x2 sequence length is too short to let it).

- L654: as you are applying in this case DNA-based methodologies, please eliminate any reference to express, as you are not evaluating how the are expressing, you are evaluated the abundance of their DNA (totally different).

Minor considerations:

Abstract: please change "Bifidobacteria" to "bifidobacteria".

L45: delete ","

L47: include "potential" instead of "possible"

L48: delete "and"

L57: change "systems" to "system"

L62 "do " Lactobacillus sp., Bifidobacterium sp. and Bacteroides sp" mean genera  Lactobacillus, Bifidobacterium and Bacteroides, or their species? if the second is the case, please change to  Lactobacillus spp., Bifidobacterium spp. and Bacteroides spp. If not, change to "genera  Lactobacillus, Bifidobacterium and Bacteroides"

L 65: please change "Bifidobacteria" to "bifidobacteria".

L71-72 Please revise the writing og “One such example is of Faecalibacterium prausnitzi which was shown to be protective against IBD in human …” and change “human” to “humans”.

L106: please change to “ad libitum

L118: please include µ symbol

L118: the parenthesis are not closed properly.

L118: does % w/vol? please cited

L125: please change “till” to “until”

L131-132 please correct writing “Disease activity index scoring was done based on the total of the weight loss score, stool consistency and stool bleeding “

L139-141: please revise the units in %; w/v, v/v?

L154: please change “ex vivo” to « ex vivo”.

L162: please identify RPMI, with its name and brand. Is it also ThermoFisher Scientific?

L164-165. units in %

KL166: change CO2 to CO2

L168: the parenthesis are not closed properly.

L178: which was the kit used for RNA isolation?

L179: please cite correctly the kit and brand

L195: please change “primers” to “primers”

Table 1. are they 5´to 3´?

L196 please change to “The genes and nucleotide sequences”

L298-299 please change Bifidobacteria to bifidobacterial

L507-510. Please revise English and writing.

L511 what is “The Bifidobacterium strain?” as you are working with 16s rRNA sequencing, you are not seeing strains. You must talk about genus, family, phylum, etc

L513-520. It is a mistake to talk about “strain”. 16s rRNA can differentiate to genus level, and hardly ever species. Only with metagenomics and total DNA analysis you will be able to discriminate at strain level. Please refers to OTU, or genus, or species.

L575: change to bifidobacteria (non italics)

L549: change to bifidobacteria

L600: rewrite “Together this suggest that curdlan was able to dampen innate immune responses in mice.”

L604: correct to “enters”

L612 correct to “lactobacilli and bifidobacteria…. And do the same in all the document. You can say Bifidobacterium referring to the genus, or bifidobacteria referring to the cells, but not mixed both terms.

L619: correct to “B. animalis subsp. lactis strain BB12”

Bifidobacteria

Reviewer 3 Report

The paper nicely describes the role of dietary curdlan to enhance Bifidobacteria in intestinal inflammation. However, several errors need to be corrected.

  1. Several typos such as 1mg, 4mg/ml, missing commas, hyphens, were observed in the manuscript. Correct them.
  2. No information about total colon length in DSS-fed mice. Can researchers include that result too?
  3. A very little description was included in the discussion section regarding comparisons of this study with other related studies. Several studies were already conducted on the related topic. Therefore, could you include more information in the discussion section?
  4. Can you add more about the significance of this study in the introduction? Is this study concluded?